# Dietary Supplementation of Microbial Dextran and Inulin Exerts Hypocholesterolemic Effects and Modulates Gut Microbiota in BALB/c Mice Models

**DOI:** 10.3390/ijms24065314

**Published:** 2023-03-10

**Authors:** Iqra Jawad, Husam Bin Tawseen, Muhammad Irfan, Waqar Ahmad, Mujtaba Hassan, Fazal Sattar, Fazli Rabbi Awan, Shazia Khaliq, Nasrin Akhtar, Kalsoom Akhtar, Munir Ahmad Anwar, Nayla Munawar

**Affiliations:** 1Industrial Biotechnology Division, National Institute for Biotechnology and Genetic Engineering College, Pakistan Institute of Engineering and Applied Sciences (NIBGE-C, PIEAS), Faisalabad 38000, Pakistan; 2Department of Microbiology, Abbottabad University of Science and Technology, Havelian, Abbottabad 22020, Pakistan; 3Health Biotechnology Division, National Institute for Biotechnology and Genetic Engineering College, Pakistan Institute of Engineering and Applied Sciences (NIBGE-C, PIEAS), Faisalabad 38000, Pakistan; 4Department of Chemistry, College of Science, United Arab Emirates University (UAEU), Al Ain 15551, United Arab Emirates

**Keywords:** dextran, inulin, gastrointestinal microbiota, enteric pathogens, prebiotics, probiotics, exopolysaccharides, hyperglycemia, hypercholesterolemia

## Abstract

Microbial exopolysaccharides (EPSs), having great structural diversity, have gained tremendous interest for their prebiotic effects. In the present study, mice models were used to investigate if microbial dextran and inulin-type EPSs could also play role in the modulation of microbiomics and metabolomics by improving certain biochemical parameters, such as blood cholesterol and glucose levels and weight gain. Feeding the mice for 21 days on EPS-supplemented feed resulted in only 7.6 ± 0.8% weight gain in the inulin-fed mice group, while the dextran-fed group also showed a low weight gain trend as compared to the control group. Blood glucose levels of the dextran- and inulin-fed groups did not change significantly in comparison with the control where it increased by 22 ± 5%. Moreover, the dextran and inulin exerted pronounced hypocholesterolemic effects by reducing the serum cholesterol levels by 23% and 13%, respectively. The control group was found to be mainly populated with *Enterococcus faecalis*, *Staphylococcus gallinarum, Mammaliicoccus lentus* and *Klebsiella aerogenes.* The colonization of *E. faecalis* was inhibited by 59–65% while the intestinal release of *Escherichia fergusonii* was increased by 85–95% in the EPS-supplemented groups, respectively, along with the complete inhibition of growth of other enteropathogens. Additionally, higher populations of lactic acid bacteria were detected in the intestine of EPS-fed mice as compared to controls.

## 1. Introduction

Gastrointestinal microbiota is recognized as a “functional organism” as it provides various immunological, defensive, structural, and metabolic benefits to the host as a result of host-microbial interaction by creating an efficient ecosystem and ensuring body homeostasis [1,2]. The pattern of the gut microbiome in different individuals is influenced by age, genetics, medication, infection, personal hygiene, allergens contact, and most importantly diet and the use of probiotics and prebiotics [3,4,5]. Probiotics and prebiotics collectively fall in the category of functional foods. Food is considered functional if along with providing nutritional value, it proves beneficial for the well-being and health of an individual and reduces the risks of diseases [6]. The functionality of functional foods is constituted by certain bioactive components present in them. Among these components, microbial exopolysaccharides (EPSs) such as fructans and α-glucans have gained much research focus in terms of determining their role in improving host health.

Fructans and α-glucans are homopolysaccharides comprising of single type of monomer, i.e., fructose or glucose, respectively. The diversity in these polysaccharides can be attributed to the fact that the constituent monomeric sugar units are linked through different types of glycosidic bonds in the backbone structure with varied branching patterns. Based on glycosidic linkage types, fructans are categorized into two major groups. These include levan and inulin, having β-(2→6) and β-(2→1) linkages among their main chain glycosidic units, respectively, with varying degrees of branching. Similarly, α-glucans are classified as dextran having α (1→6) linkages in the main chain with α-(1→3) linked branches, mutan having α-(1→3) linkages with α-(1→6) linked branches, reuteran having α-(1→4) linkages and alternan with α-(1→6)/α-(1→3) linkages between the D-glucose monomers. Fructans are present in bacteria, fungi, archaea and plants [7,8] while α-glucans are exclusively produced by lactic acid bacteria (LAB) [9].

Prebiotic exopolysaccharides (EPSs) are not hydrolyzed by digestive enzymes of humans and non-ruminant animals lacking carbohydrate active enzymes (CAZymes), but undergo bacterial fermentation in the distal part of gut by the CAZymes of the gut microbiome [10,11]. They interact with mucosa and microflora to regulate “gut health”. The enrichment of the population of advantageous bacteria by fermenting prebiotic exopolysaccharides not only enhances their ability to compete with the potential pathogens for the available nutrients but also produce short chain fatty acids that inhibit pathogen proliferation by lowering the pH of the lumen leading towards the competitive exclusion of pathogens by preventing their attachment to epithelial cells [12]. Hence prebiotics enhance the growth of beneficial bacteria that exhibit quorum sensing, competition for resources, adherence properties and possession of different metabolic pathways, leading to the eradication of the less desirable members of the intestinal microflora [13,14].

Besides the above-mentioned beneficial aspects of polysaccharides on the gut microbiome, a few studies have reported their impact on health metabolomics indicators and biochemical parameters, such as blood glucose and/or cholesterol levels. Notably, some investigators reported a significant reduction in serum cholesterol concentration in inulin-fed mice [15,16] while others observed no significant difference in serum cholesterol levels between inulin-fed model organisms (rats or mice) and controls [17]. It may be noted that mostly inulin extracted from plants has been used in these studies. A few investigators have used microbial levan in their studies and reported its cholesterol lowering effects on rats [18,19]. Thus, paucity of data on health beneficial effects of microbial EPSs and contradictory results accentuates further experimentation in this direction. In this paper, we have compared the effects of two different types of bacterial EPS, i.e., inulin and dextran, on serum glucose and cholesterol levels of mice. Further, we have also investigated their effects on the populations of culturable lactic acid bacteria and some common enteric pathogens in the mice gut. To the best of our knowledge, this is the first in vivo study on microbially produced EPSs to elucidate their metabolomic and microbiomic impacts in the gastrointestinal tract using mice models.

## 2. Results

### 2.1. Microbial Synthesis of Exopolysaccharides

Both polymers were synthesized under already optimized conditions to acquire the maximum yield. In physical appearance, inulin is non-sticky and has a white to pale tint whereas dextran has a white color and a sticky texture. Both of these polymers were obtained in fine powder form after freeze drying. In order to avoid the presence of any residual sucrose, the purity status of the synthesized polymers was confirmed by thin layer chromatography. After developing the TLC plates with urea developing solution (for inulin) and methanol developing solution (for dextran), no degradation or additional spots were observed on the TLC plate except that of polymers, confirming that polymers were in pure intact form and free of other substances such as residual sucrose and glucose (Figure 1).

### 2.2. Body Weight Analysis

As already mentioned, the body weights of all the mice of three groups were determined on Day 0 and Day 21 of the experiment. Figure 2 shows the weight gain as percentage of their initial body weights during the course of this experiment on supplementation with EPS (5% *w*/*w*) in their feed. The initial weights among the groups were not significantly different. After 21 days of feeding, 19 ± 2.3% gain in total body weight was observed in the control group. On the contrary, only 7.6 ± 0.8% weight gain was observed in the inulin fed mice group. Though weight gain in the dextran group was not considered statistically significant (*p* < 0.05), it showed a lower (15.7 ± 1.8%) trend in comparison with the control group.

### 2.3. Blood Glucose Analysis

Blood glucose levels of the control group of mice showed a significant increase (22 ± 5%) over a period of 21 days (Figure 3). Contrarily, no significant change in the blood glucose levels of the experimental groups was observed after 21 days of feeding on dextran- or inulin-supplemented feed, as compared to those on Day 0. Further, the blood glucose levels of both experimental groups were found to be significantly lower than those of the control group at the end of the experiment (day 21).

### 2.4. Serum Cholesterol Analysis

Serum cholesterol levels of mice from all the diet groups were measured on the day of euthanasia (i.e., day 21). It was observed that after feeding the mice with inulin- or dextran-supplemented diet, their cholesterol level was reduced as compared to the control group. Dextran exhibited a more pronounced hypocholesterolemic effect as it reduced the serum cholesterol level by 23% while inulin reduced it by 13%, compared to the control group (Table 1).

### 2.5. Fecal Lipid Analysis

Analysis of total lipids in the fecal samples collected during the last 48 h of the experiments showed that feeding the mice with dextran had no significant effect on the amount of excreted lipids as compared to the control group (Table 1). However, feeding with inulin-supplemented feed resulted in a profound increase (about 78%) in the concentration of total lipids excreted in feces, in comparison with the samples from the control group.

### 2.6. Organ Weight Analysis

After euthanasia of the mice on day 21 of the experiment, weights of their different body organs were determined. These organs included liver, kidneys, pancreas, lungs, heart, stomach and spleen. Comparison of the weights of these organs from inulin- and dextran-fed groups with the control group showed no significant difference (Table 2).

### 2.7. Analysis of Intestinal Microbiota of Mice

Multi-colored colonies with various morphological appearances were obtained after a 16-h incubation period at 37 °C, when the intestinal fluid of mice from different treatment groups was spread on CHROMagar plates. The CHROMagar color orientation provided by the manufacturer was used for preliminary identification of the colonies. CHROMagar colony data showed that the microbiota in the intestinal fluid of the control group of mice fed with normal diet were mainly populated by *Enterococcus* (turquoise blue colored colonies) along with a significant population of *Klebsiella* (metallic blue colored colonies), *Staphylococcus* (cream-colored colonies) and *Mammaliicoccus* (sky blue colored colonies) species. A few pink colonies that belonged to *Escherichia* sp. were also seen in the intestinal microbiota of control mice. In contrast, microbiota in the intestinal mucosa (scraped material) exhibited a significantly higher population of *Escherichia* sp. along with all other bacteria found in the samples of the same control animals, raising the likelihood of the attachment of *Escherichia* sp. to intestinal epithelial cells.

Interesting differences were found in the microbiota of dextran and inulin treatment groups of mice. The CHROMagar plates inoculated with intestinal fluid of dextran and inulin fed mice were dominantly populated with *Escherichia* sp. along with significantly lower *Enterococcus* population as compared to intestinal microbiota of the control group. No additional colored colonies were seen in the microbiota of both experimental groups of mice, indicating that feeding on the inulin- or dextran-supplemented feed had inhibited the growth of *Klebsiella*, *Mammalicoccus* and *Staphylococcus* species compared to the microbiota of control group (Appendix A).

The role of microbial dextran and inulin in the eradication of *Escherichia* sp. was further confirmed by the results of MacConkey agar analysis which showed the similar microbial pattern as exhibited by CHROMagar analysis. It was observed that the plates dispersed with intestinal fluid of the control group were predominantly occupied by red and pale-pink colored colonies indicating the presence of *Enterococcus* and *Staphylococcus*, respectively. In contrast, the plates spread with intestinal fluid of dextran and inulin treatment mice groups were predominantly occupied by non-mucoid pink colored colonies indicating the presence of *Escherichia* species (Appendix A). These results strengthened the evidence that microbial dextran and inulin could play a significant role in the detachment of *Escherichia* sp. from intestinal epithelial cells and enhance the eradication of intestinal pathogenic bacteria.

In addition to the eradication of pathogens, our quest was to investigate the effect of microbial dextran and inulin supplementation on the proliferation of beneficial microbiota in the gastrointestinal tract of mice. For this purpose, the intestinal fluids of all groups of mice were spread on MRS-agar plates supplemented with 2% glucose or 20% sucrose. After overnight incubation, a smaller number of colonies appeared on the plates of the control group but a significant population was observed on the plates spread with intestinal fluid of dextran- or inulin-fed groups of mice. Some results were surprisingly different in the intestinal fluid of the experimental group of mice analyzed on MRS-glucose plates as we observed large milky white colonies on these plates. These colonies were further grown and purified in YPD (yeast extract peptone dextrose) medium supplemented with kanamycin. When observed under the microscope, these white colored colonies appeared to be composed of yeast cells (Appendix A). These results indicated that the exopolysaccharides used in this study could have a prebiotic role exhibiting strong potential to increase the population of beneficial microbiota by reducing the pathogenic bacterial species.

### 2.8. Molecular Identification and Phylogenetic Analysis of Intestinal Microbial Isolates

All morphologically distinct colonies from each plate were selected, purified and subjected to DNA extraction followed by 16S rRNA gene sequencing. Each bacterial colony’s 16S rRNA gene was PCR amplified using FD1 and RP1 primers, producing a 1500 bp-sized product. The Blastn program, offered by NCBI, was used to analyze the sequencing data of the PCR-amplified genes and reveal their percentage identity with other closely related strains. According to these results, the turquoise blue colony (labelled as IJ4) from the control group exhibited 100% identity with *Enterococcus faecalis* strain NBRC 100481 whereas the golden opaque (labelled as IJ5), light blue (Labelled as IJ9) and metallic blue (labelled as IJ11) colored colonies from the control group showed 99.8%, 99.9% and 99.68% identity with *Staphylococcus gallinarum* strain VIII1, *Mammaliicoccus lentus* strain MAFF 911385 and *Klebsiella aerogenes* strain NBRC 13534, respectively. The Blastn of the 16S rRNA gene for pink colored colonies from experimental groups showed that these colonies were 99.49% identical with *Escherichia fergusonii* strain NBRC 102419. This homology was further confirmed through phylogenetic analysis by phylogenetic tree that was constructed using the MEGA11 program employing neighbor joining algorithm. All of the isolates share their cluster with most identical strains as depicted by Blastn results (Figure 4).

Using 16S rRNA gene sequence homology, isolates from MRS agar plates inoculated with intestinal fluids from dextran and Inulin-fed groups were also identified. According to Blastn results, the intestinal fluid of these mice contained three different types of bacteria in a dominant proportion labeled as IJ6, IJ7 and IQ4 having 98.11%, 99.80% and 98.64% homology with *Lactococcus garvieae* strain NIZO2415T, *Bacillus subtilis* strain BCRC 10255 and *Bacillus licheniformis* strain ATCC 14580, respectively. The phylogenetic analysis revealed that these microbes cluster closely with the above said respective closely related bacterial species (Figure 5). All the 16S rRNA gene sequences obtained in this study were submitted to NCBI and their accession numbers were obtained as given in Table 3.

### 2.9. Comparative Analysis of Intestinal Microbial Populations among Different Treatment Groups

The comparative analysis of culturable microbial populations among different treatment groups showed that the population of *Enterococcus* was significantly inhibited down to 59% and 65% in intestinal fluids of dextran and inulin-fed mice, respectively, as compared to the control. Similarly, the release of intestinal *E. fergusonii* population in dextran and inulin-fed mice enhanced their levels up to 89% and 95%, respectively, as compared to the control. The occurrence of a significant population of *E. fergusonii* in intestinal mucosa samples of the control also showed that both polymers trigger the detachment and release of *E. fergusonii* from the intestinal epithelial cells. The populations of *K. aerogens*, *S. gallinarium* and *M. lentus* were completely inhibited in the experimental groups demonstrating the effectiveness of microbially derived exopolysaccharides in maintaining gastrointestinal health by inhibiting the growth of enteric pathogens (Figure 6). The statistical analysis also revealed a significant difference in the number of colony forming units (CFU) of *Enterococcus* and *E. fergusonii* in dextran and inulin-fed groups as compared to the control (Table 4).

## 3. Discussion

In spite of considerable information available on the synthesis and characterization of microbial EPSs, their experimental usage, especially in regard to biomedical applications, is limited only to plant-based inulin, fructo-oligosaccharide and some β-glucans. Recently, there has emerged a great deal of interest in the ability of EPSs of microbial origin to confer potential health benefits in animals and humans. In this context, levan type polysaccharide of bacterial origin has been shown to exert hypocholesterolemic effects in experimental mice [19]. In the present study, we have used two different types of microbial EPSs, i.e., dextran and inulin, in experimental mice models to investigate their hypoglycemic and hypocholesterolemic potentials.

The experimental mice were fed with 5% (*w*/*w*) EPSs, as this concentration has been found to be optimum for a noticeable hypocholesterolemic effect in the previous study by Yamamoto et al. [19]. Considerably less weight gain was observed in the mice fed on the EPS-supplemented diets in comparison to the control group mice. These results are contradictory to those reported by Yamamoto et al. [19], where levan was used as diet supplement for rats. In that study, no significant differences in the body weight gain were found between levan-fed rats and the control rats over the experimental period. Our results support the fact that upon consumption of prebiotic EPSs in their diet, the body weight gain of mice was controlled. This control in body weight could be explained on the basis of some previously reported mechanisms. One such mechanism is the potential reduction in the amount of calories obtained via fermentation of these EPS and the consequent absorption and metabolism of their fermented products, i.e., short chain fatty acids (SCFAs) [20]. The calories obtained are reported to be of less than half of the caloric value gained from direct metabolism of a similar amount of carbohydrates [21]. Some studies suggest a more complex phenomenon for this reduced caloric value by highlighting the variation of gut microbiota and its influence on harvesting energy from EPS [22]. It is also a well-known concept that differences in dietary patterns could lead to variation of the gut microbiota [23]. Thus, it can also be assumed that in the present study, mice fed with EPS developed variations in their microbiota composition compared to the control group and the resulting gut microbial diversity could be responsible for influencing dietary intake (satiety) and energy expenditure. Another explanation for this appetite regulation is the interaction of SCFAs with free fatty acid receptors FFA2 and FF3 which leads to the varied expression of hormones and peptides, influencing the energy metabolism [24].

Many natural polysaccharides are known to be potential anti-diabetic agents [25]. A majority of polysaccharides with anti-diabetic activity have yet not been validated scientifically due to the complexities in their structure and their undefined mechanisms. Further research on the potential anti-diabetic mechanisms of polysaccharides is the need of the hour. In the present studies, both the inulin and dextran were found to exert significant hypoglycemic effects on mice. In these studies, no significant change in the blood glucose level of inulin- and dextran-fed mice was observed over the experimental time span. One possible explanation for this hypoglycemic effect could be that the EPSs shorten gastric emptying time and reduce the transit time of the small intestine which in turn prevents postprandial elevations of blood glucose [26]. Another possible mechanism of controlling blood glucose level could be the influence of SCFAs on host metabolism as stated earlier. For instance, propionate has been found to upregulate hepatic glycolysis and downregulate hepatic gluconeogenesis. Similarly, butyrate in the colonic lumen enhances the expression of an anti-hyperglycemic hormone called glucagon-like peptide-1 (GLP-1) which is believed to promote pancreatic beta cell proliferation, thus increasing the amount of insulin secretion [27]. Further broad investigations on human subjects are required for the validation of these studies and to determine the exact mechanism of action.

Lower cholesterol levels, as compared to the controls, were observed in the blood sera of the mice fed with inulin and dextran supplemented feed. Between these, dextran exhibited a more pronounced effect on lowering the serum cholesterol level as compared to inulin. These results indicate that inulin and dextran have the potential to exert hypocholesterolemic effects. It has been documented that inulin and other fermentable soluble dietary fibers exert hypocholesterolemic effects by two mechanisms, i.e., selective fermentation by intestinal microbiota leading to the production of SCFAs, and enhanced cholesterol excretion through feces due to decreased cholesterol absorption [28]. It has also been proposed that the fermentation products, especially propionate and other SCFAs might regulate cholesterol metabolism. Reabsorption of bile acid is reduced and colonic pH is reduced because of the production of SCFAs [29]. Moreover, propionate may also affect liver cholesterol synthesis by reaching the liver [30]. Not all investigators agree to such roles of SCFAs. Hence, there is no concrete evidence of SCFAs role in lipid metabolism, particularly for propionate [31]. However, there could be a link between serum cholesterol level and lipid content excreted through feces. Indeed, the binding and entrapping properties of dietary fibers have been found to result in lower levels of steroid re-absorption [32]. Similarly, in the previous study by Yamamoto et al. (1999) [19], levan-fed rats have been found to have remarkably higher amount of lipids and total sterols in their fecal excretions, as compared to controls. These results suggested that levan was responsible for disturbing re-absorption by binding and entrapping the steroids in the intestine. In the present study also, inulin was found to have a very pronounced effect on the fecal lipid concentration, where it increased to 78% in comparison to the control. However, dextran had no effect on the concentration of lipids excreted through the fecal matter, though feeding the mice on a dextran-supplemented diet drastically reduced their serum cholesterol level. Inulin might have exerted this hypocholesterolemic effect via the same mechanism of binding and entrapping lipids as reported in the previous studies with other dietary fibers, whereas some different mechanism operates behind the hypocholesterolemic effect of dextran, which needs further exploration.

Another aspect of the present study was determining the changes in certain intestinal microbial populations induced in mice models fed on dextran and inulin supplemented diets. The balance between the commensal microflora, the diet and the mucosa are vital to maintain the gut health while any imbalance or dysbiosis could lead to adverse health effects including inflammatory bowel disease (IBD), multiple sclerosis, Alzheimer’s disease, diarrhea, colorectal cancer, etc. [33]. In this context, studies on the impact of prebiotic exopolysaccharides on a variety of health-promoting, pathogen-inhibiting, and microbiota-modulating activities are being undertaken, both in vitro and in vivo.

The key findings of our study highlighted that both of the tested microbial exopolysaccharides i.e., dextran and inulin, cause a shift in the gastrointestinal microbiota through drastic reduction in the proliferation of *Enterococcus faecalis*. These polymers were also found highly effective against many other common enteric pathogens including *Enterococcus*, *Klebsiella*, *Staphylococcus* and *Mammaliicoccus* species that were found as major inhabitants in the gastrointestinal tract of the control group of mice fed with a normal diet. In previous investigations, the enteric pathogen inhibitory potential of exopolysaccharides was mostly investigated in vitro. In contrast, our study is among very few in vivo investigations using mice models to evaluate the inhibition of enteric pathogens in response to the dietary supplementation of microbial dextran and inulin type EPSs. The available literature shows that EPSs exhibit strong enteropathogen inhibitory potential. For instance, the EPS released from *Lactobacillus acidophilus* A4 was found to decrease the biofilms of *E. coli* (EHEC) by 87% along with the inhibition of Gram negative and Gram positive pathogens [34]. In another study, it was found by Osamu Kanuchi et al. (2002) that the administration of prebiotic germinated barley foodstuff (GBF) enhanced the population of bifidobacteria and *Eubacterium limosum* while inhibiting or reducing Bacteroides, leading to improvement in the ulcerative colitis in patients [35]. Previously, Xiaoqing Xu et al. (2020) found that *Lacticaseibacillus casei* NA-2 derived EPS possesses strong growth and biofilm inhibitory potential against *Staphylococcus aureus*, *Bacillus cereus*, *E. coli* O157:H7 and *Salmonella typhimurium* exhibiting strong inhibition ratios of 30.2% ± 3.3%, 95.5% ± 0.1%, 16.9% ± 5.4%, and 14.3% ± 0.6%, respectively, which promotes the antibacterial activity of *L. casei* NA-2 [36]. Our results strengthened these previous findings that prebiotic EPSs play key role in the modulation of gastrointestinal microbiota by eliminating or inhibiting enteropathogens. There may be a number of causes for this inhibition. It has also been reported that the fermentation of prebiotics leads to the enrichment of the population of advantageous bacteria which competitively exclude pathogens by competing them for available nutrients. They are reported to produce short chain fatty acids (acetate, propionate, butyrate, etc.) that inhibit pathogen proliferation by lowering the pH of the lumen leading towards the competitive exclusion of pathogens less tolerant to low pH [12,37]. Prebiotics either directly bind to pathogens or raise the intestinal lumen’s osmotic pressure. In addition, they produce different antagonistic agents such as organic acids and anti-microbial peptides for direct inhibition of pathogens [38]. All these considerations are relevant to this present study which describes the significant inhibition of common enteric pathogens in response to prebiotics supplementation.

Another interesting finding of the present study is the increase in the release of intestinal *E. fergusonii* by preventing its attachment to epithelial cells. Among nine species of *Escherichia*, *E. fergusonii* is a relatively new species discovered for the first time in 1985 from clinical blood samples, with a close resemblance to *E. coli* [39]. Its major health complications include urinary tract infections or bacteremia and open wounds infection [40]. *E. coli* is extensively used as a model organism for adhesion related investigations due to its distinctive adhesion manner. Various oligosaccharide-binding proteins also called adhesins are expressed in *E. coli* through which it binds with the oligosaccharide receptors on the surface of host cells. Since it represents the first stage of infection, the initial non-intimate adherence is a crucial feature of EPEC pathogenesis [41]. By preventing this early adhesion, the infectious process may eventually be stopped. In the past, few in vitro studies have explored the anti-adherence potential of prebiotics against *E. coli*. According to a study conducted by Zhengqi Liu et al. (2017), the administration of *Lactiplantibacillus plantarum* WLPL04 derived EPS significantly inhibited the adhesion of *Escherichia coli* HT-29 cells in competition, replacement, and inhibition assays at a dose of 1.0 mg/mL along with inhibition against biofilm formation of *Pseudomonas aeruginosa* CMCC10104, *E. coli* O157:H7, *Salmonella* typhimurium ATCC13311, and *Staphylococcus aureus* CMCC26003 [42]. In another in vitro study, galacto-oligosaccharides at an optimum dosage of 16 mg/mL were reported to inhibit the adhesion of enteropathogenic *E. coli* to the epithelial cells such as HEp-2 and Caco-2 cells by 65 and 70%, respectively, due to their structural homology with the pathogen binding sites [41].

All of these previous studies were performed in vitro using tissue cultures. In contrast, our study includes detailed in vivo analysis and molecular characterization to confirm the attachment inhibition of enteropathogens. Moreover, to the best of our knowledge, this is the first in vivo study which was performed by accessing the microbiota of mice model to describe the anti-adhesive properties of dextran and inulin against *E. fergusonii*, which is the closest neighbor to *E. coli*. The results illustrated that the intestinal fluid of control mice did not harbor significant populations of *E. fergusonii* while the intestinal mucosa exhibited their presence as a prevalent population; this confirms the hypothesis that *E. fergusonii* are attached with the intestinal epithelium and get detached when the intestine of same control animal is gently scraped with the sterile spatula. N contrast, in the intestinal fluid of dextran and inulin fed mice, the *E. fergusonii* was present as a major population, indicating that prebiotic EPSs not only significantly reduce the prevalence of enteropathogens but are also involved in promoting the eradication of *E. fergusonii* from the intestinal tract by preventing its adhesion to epithelial cells. The reason behind this may be that EPSs structurally mimic the pathogen binding sites that coat the surface of gastrointestinal epithelial cells, preventing pathogenesis and leading to the eradication or flushing out of the pathogens from the GI tract [41].

Besides the pathogenic inhibitory and anti-adhesive potential of dextran and inulin, our study was also focused on the increase in the population of beneficial microbes or lactic acid bacteria that might have been involved in the competitive exclusion of pathogens. Our results depict that the intestinal fluid of dextran- and inulin-fed mice harbor significantly higher populations of beneficial microorganisms including *Lactococcus garvieae*, *Bacillus subtilis* and *Bacillus licheniformis* when cultured on MRS-agar plates. In some previous studies, fructo-oligosaccharides (FOS) and inulin have been reported to inhibit the growth of *E. coli* and *Salmonella* in the intestine along with enhancing the host defense mechanism by boosting the metabolic activity of the lactobacillaceae family and bifidobacteria resulting in the inhibition of enteropathogens, a process called “resistance of colonization” [43]. The microbial glucans are also proved to increase stress tolerance and probiotic potential by promoting the growth of lactobacilli in the colon [44]. In addition to strengthening these previous findings, our study highlighted an interesting fact: prebiotic EPSs are also involved in enhancing the proliferation of yeast, which is a relatively new aspect of this study.

The overall gut health is influenced by the diet, mucosa and commensal flora as they ensure efficient functioning of the digestive system by interacting with host epithelial cells forming a delicate and dynamic equilibrium within the alimentary tract [45]. The intestinal epithelium is composed of diverse cells exhibiting tight junctions between them to maintain a barrier [46] that is breached by the enteric pathogens due to release of their toxins; the resulting leakage of luminal content into the lamina affects residing immune cells and instigates inflammatory responses that ultimately lead to serious health challenges [47,48]. Pathogenesis is initiated with the adherence of pathogenic bacteria to the mucosal epithelial cells which initiates ‘crosstalk’ between the microbial and epithelial cells before colonization of the colon, with continuing flow of intestinal chyme, triggering a signaling cascade to stimulate the intense inflammatory immune response. The adherence is assisted by the binding of carbohydrate binding protein receptors of pathogens with 3–5 monosaccharides long oligosaccharides receptor sites on surface of epithelial cells. Any failure to adhere leads to rapid elimination of enteric pathogens from the gut [43,49].

Besides stimulating the growth of probiotics, it is reported that structural resemblance is present between the prebiotics and these oligosaccharides receptor sites [14,49,50]. Prebiotics incorporate similar non-digestible oligosaccharides which act as blocking factors by mimicking the ligands for protein receptors, preventing the pathogen adhesion to mucosal cells [51,52]. Certain terminal sugars on these oligosaccharides, e.g., oligofructose, interfere with the receptors by binding to the bacteria and preventing attachment to the same sugar on microvillus glycoconjugates. Due to this anti-adhesive property, prebiotics work as decoy molecules for the pathogenic microorganisms leading to their displacement or flushing from the gastrointestinal (GI) tract by preventing their adherence, causing a decrease in their pathogenic potential [53,54]. In addition, prebiotics have a major advantage over antibiotics whose extensive use may result in the persistence of resistant pathogenic bacteria which may pose threats to human or animal health. Instead of using antibiotics to protect the host against the colonization of pathogenic bacteria, the most viable approach is to use dietary components that can enhance the colonization resistance against entero-pathogens [45,55]. Being safe, affordable, cheap, and accessible, prebiotics are best candidate for this purpose [56].

## 4. Materials and Methods

### 4.1. Production and Purification of Exopolysaccharides

Two different kinds of microbially produced exopolysaccharides, i.e., dextran and inulin, were used in this study. Dextran was produced by a high yielding *Weisella ciberia* strain M3b. This strain has been previously isolated from fermented malt grains and identified by 16S rRNA gene sequence analysis that has been submitted to NCBI under the GenBank accession number MH084846 (data to be published separately). This isolate has been deposited to NIBGE Biotech Resource Centre (NBRC) (http://nibge.org/Division.aspx?page=Plant%20Microbiology&div=EBT), Pakistan, under the accession number NBRC-528 with an open access to other researchers on payment of shipment charges. Inulin was produced by *Lactobacillus gasseri* DSM 20604 strain. For EPS production, the cultures of both of these bacteria were grown in De Man, Rogosa and Sharpe agar (MRS) medium having the composition: Peptone (Rapid Labs, Cholchester, Essex, UK) 10 g/L, beef extract (Bio Basic Inc., Konrad Crescent Markham, ON, Canada) 8 g/L, yeast extract (Biochem, Za Conse Sur Ioire, France) 4 g/L, K_2_HPO_4_ (MP Biomedicals Inc., Solon, OH, USA) 2 g/L, sodium acetate (Sigma Aldrich, Saint Louis, USA) 3 g/L, diammonium citrate (Uni Chem, Mumbai, India) 2 g/L, magnesium sulphate (Riedel-de Haen AG Seelze, Hannover, Germany) 0.2 g/L, manganese sulphate (Sigma Aldrich, Saint Louis, MO, USA) 0.5 g/L, Tween 80 (Bio Basic Inc., Markham, ON, Canada) 1 mL/L, pH 6.2–6.5) [57] supplemented with 20% sucrose (PhytoTechnology Laboratories, Shawnee Mission, KS, USA). The *W. cibaria* and *L. gasseri* cultures were incubated at 28 °C under aerobic conditions and 37 °C under anaerobic conditions, respectively, for 96 h to get the maximum possible EPS yields. Presence of exopolysaccharides in the cultures and purity of the purified polysaccharide products were determined through thin layer chromatography (TLC) analysis. For this purpose, 2 μL of the sample was run on TLC plates (Silica gel 60 F_254_; Merck, Darmstadt, Germany) for about 6 h in a mobile phase comprising butanol/ethanol/water (5:5:3). To detect the fructose containing carbohydrates (e.g., inulin), TLC plate was air-dried and sprayed with urea developing solution (100 mL water saturated butanol, 3.0 g urea, 5.9 mL phosphoric acid, 5 mL ethanol) followed by heating at 120 °C for 15 min [58]. Other carbohydrate spots (e.g., dextran) were visualized by developing with a solution containing 5% sulfuric acid in methanol.

In order to purify EPSs, the *W. cibaria* and *L. gasseri* cultures were centrifuged at 4824× *g* for 20 min to remove the bacterial cells. The supernatants containing the EPSs were collected and the cell pellets were discarded. The proteins present in the cultures were removed by treatment with tricholoroacetic acid (TCA) following the method described by Abid et al. (2019) [59]. The dextran polymer was precipitated from the TCA-treated culture supernatant by adding one volume of ice-cold absolute ethanol, while two volumes of ethanol were used for the precipitation of inulin. The precipitated polymers were then washed by dissolving in distilled water and centrifuging again at 4824× *g* for 10 min. The pellets were preserved while the supernatants were discarded. This step was repeated 2–3 times until the purified polymers were obtained. Both the polymers were then dissolved in distilled water and analyzed by thin layer chromatography (TLC) to confirm the absence of any residual sucrose and glucose molecules. Finally, the polymers were suspended in water and freeze-dried to obtain in fine powder form.

### 4.2. Experimental Design

Total 18 male BALB/c mice aged five weeks were purchased from Government College University Faisalabad, Pakistan, and equally divided into three groups i.e., control group, dextran-treated group, and inulin-treated group. Before starting the experiment, all the mice were given normal control diet in order to acclimatize them to the new environment. All three groups of mice were kept in their respective cages placed in a room where temperature was maintained at 27 ± 1 °C. Each group of mice was placed in a separate cage. The control group was only fed with a normal diet with no prebiotic polymer added while the other two experimental groups were fed with the diet mixed with the inulin and dextran polymers (5% *w*/*w*). Mice were kept under observation for 21 days. Daily feedings were given in accordance with their needs and the animals were given free access to pure water and meals.

The diet was prepared by mixing purified powdered forms of prebiotic exopolysaccharides with commercially available mice feed. The feed was sterilized in an autoclave at 121 °C for 10 min to ensure the absence of any exogenous bacteria prior to the addition of the polymers. The feed was prepared on daily basis to avoid any contamination and decay.

### 4.3. Animal Euthanization and Sample Collection

Feed was removed about 20 h before the end of the experimental period [19], after which the mice were anesthetized with pentobarbital using standard ethical procedures. After waiting for 5 to 10 min until mice were completely unconscious, they were euthanized. Intestines were separated and squeezed to collect the intestinal fluid sample of all groups of mice. In addition, the intestine of the mice was also scraped gently to collect the intestinal mucosa from the walls of the intestine. The fluid samples were suspended in 10 mL of sterilized normal saline (0.9% *w*/*v* NaCl (Biochem, Za Conse Sur Ioire, France) and mixed gently to homogenize it by vortexing at low speed. Samples were labelled and stored at 4 °C for further analysis.

All animal-related procedures were according to the guidelines of the Animal House Committee after approval from the institutional biomedical ethical committee. International Guiding Principles for Biomedical Research Involving Animals as issued by the Council for the International Organizations of Medical Sciences were followed in these experiments.

### 4.4. Determination of Body and Organ Weight

Overall body weights of mice of all three groups of this study were determined at the beginning (Day 0) and the end of experiment (Day 21) a using top load balance. The mice having similar initial body weights were selected for use in these experiments. The weights of different body organs were also measured at the end of the experiment using an analytical balance.

### 4.5. Analysis of Fecal Lipids

Beddings in all the mice cages were replaced with fresh material, 48 h before the end of experiments. At the last day of the experiment, feces were collected from the cages for the determination of total lipids excretion. The fecal samples were lyophilized, ground and stored at −20 °C for further analysis. Total lipids were extracted from 1.0 g of fecal sample from each experiment in triplicate following the protocol described by Kraus et al. (2015) [60] and measured gravimetrically.

### 4.6. Blood and Tissue Collection

At the end of the experimental period, the animals were anesthetized with pentobarbital and euthanized. The blood was collected by heart puncture and instantly transferred to Gel & Clot Activator tubes (Xinle, China) and kept on ice. The tubes were centrifuged at the speed of 3000 rpm for 15 min at 0 °C. Serum was separated and preserved in separate tubes at −20 °C. After blood sampling, the heart, kidneys, liver, lungs, pancreas, spleen and stomach were detached and weighed after drying with paper tissue.

### 4.7. Glucose and Cholesterol Analysis

At the beginning of the experiment (day 0) and at the end of the experiment (day 21), blood drops were collected from each mouse by puncturing the vein in the tail with a sterile syringe needle. The concentration of blood glucose was determined using On Call Extra blood glucose monitoring system (ACON Laboratories, Hannover, Germany). Cholesterol levels in the preserved blood sera (from Section 4.6) were measured by Microlab 300 analyser (Merck, Darmstadt, Germany).

### 4.8. Analysis of Intestinal Microbiota

In order to examine the presence and diversity of common enteric pathogens, intestinal microbiota of mice were examined using CHROMagar (Kanto Chemical Company Inc., Tokyo, Japan) and MacConkey agar media. Following the manufacturer’s instructions, the media were prepared by adding 33 g and 49.53 g of CHROMagar and MacConkey agar powder respectively to 1 L of distilled water followed by sterilization at 121 °C. Using sterilized normal saline, all samples containing intestinal fluid and mucosa from control and experimental animals were serially diluted 100–1000 times. 100 µL of each dilution was dispersed on both media agar plates and overnight incubation was provided at 37 °C to allow different microorganism to grow.

To study the presence and abundance of lactobacilli in the colon of the control and experimental group of mice, all samples were similarly dispersed on MRS agar plates supplemented with 20% sucrose and others with 2% glucose. After overnight incubation, multicolored colonies were enumerated and compared in order to determine their relative abundance in control and experimental groups.

### 4.9. Isolation and Molecular Characterization of Various Intestinal Isolates

Different colored colonies which appeared on the CHROM agar and MRS agar plates were isolated, purified and primarily characterized by phase contrast microscopy for initial confirmation of the morphology and the presence of similar microorganisms indicated in CHROMagar orientation. These isolates were further allowed to grow in LB media for genomic DNA extraction whereas the isolates from the MRS agar plates were grown in MRS broth. Some large white color colonies were grown on YEPD (yeast extract peptone dextrose) medium (Peptone 20 g/L, yeast extract 10 g/L, dextrose 20 g/L) supplemented with kanamycin. Genomic DNA of all these isolates was extracted by using Thermo-scientific Gene-jet Genomic DNA extraction kit.

For molecular identification, genomic DNA of all the isolated microbes was used as template for their respective 16S rRNA gene amplification by using universal primers FD1 (forward primer) and RP1 (reverse primer) as described previously [61]. PCR products were purified through GeneJet PCR purification kit by following user provided instructions and commercially sequenced by Macrogen, Republic of Korea, using Sanger Sequencing method. Homology of the sequences was determined through the NCBI blastn program. The sequences of closest neighbors were retrieved from GenBank and phylogenetic trees were constructed by the neighbor joining method using MEGA11 (version 11.0.10) software.

### 4.10. Statistical Analysis

Graph-Pad Prism (Version 5) was used to analyze the statistical significance of data. One-way analysis of variance (ANOVA) was performed and the mean differences among the different treatment groups were evaluated. Tukey’s multiple comparison test was applied to analyze the statistical differences among means (*p* < 0.05).

## 5. Conclusions

In view of all the above-mentioned findings, it can be concluded that prebiotic EPSs are highly effective ingredients not only for improving metabolomics but also for the modulation of gastrointestinal microbiota in a beneficial manner. Besides exhibiting strong *Enterococcus* inhibitory potential, this study demonstrated the anti-adhesive properties of prebiotics against *E. fergusonii* to prevent its pathogenesis and strengthened the evidence that prebiotic EPSs promote the growth of beneficial bacteria and yeast in the GI tract. All these findings declare these EPSs as suitable candidates for use as prebiotics.

## Figures and Tables

**Figure 1 ijms-24-05314-f001:**
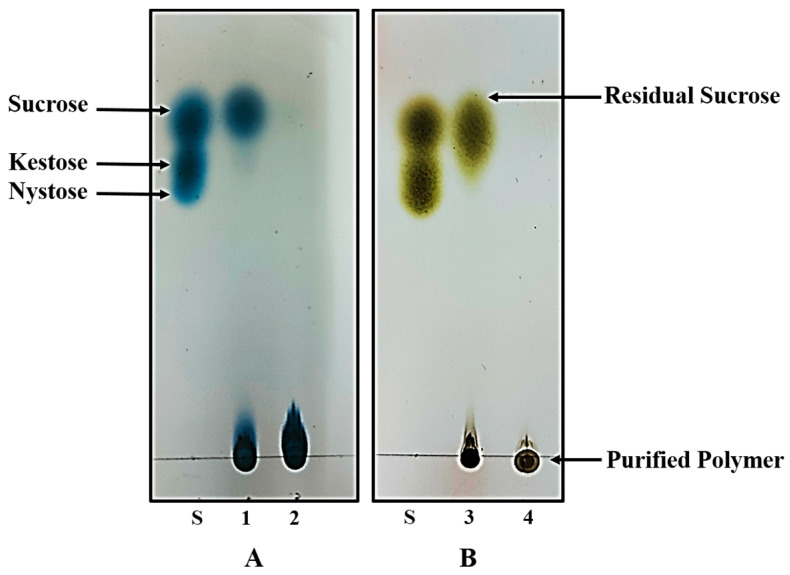
Thin layer chromatography analysis of the exopolysaccharide products synthesized by *L. gasseri* DSM 20604 (**A**) and *W. cibaria* M3b (**B**). 1: Inulin containing broth of *L. gasseri* culture; 2: Purified inulin. 3: Dextran containing broth of *W. cibaria* culture; 4: Purified dextran.

**Figure 2 ijms-24-05314-f002:**
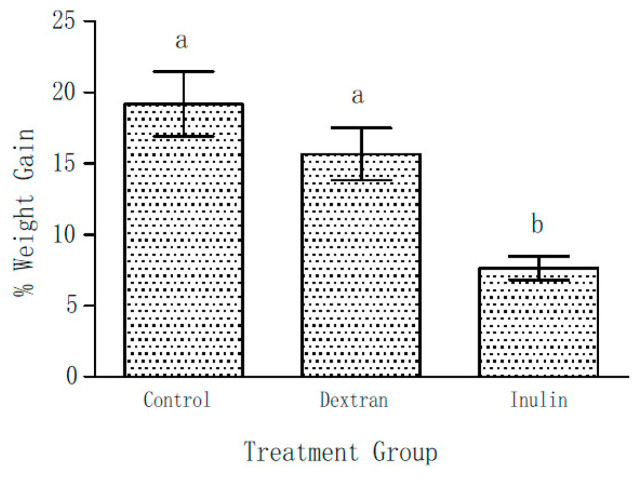
Weight gain shown as percentage increase over the initial (day 0) body weight after 21 days of feeding the mice with and without EPS supplemented feed. Data represent mean ± SEM. (*N* = 6 for each group). The bars labelled with the same letter are not significantly different i.e., ^a^
*p* > 0.05; ^b^
*p* < 0.05 compared with the control and dextran-fed groups.

**Figure 3 ijms-24-05314-f003:**
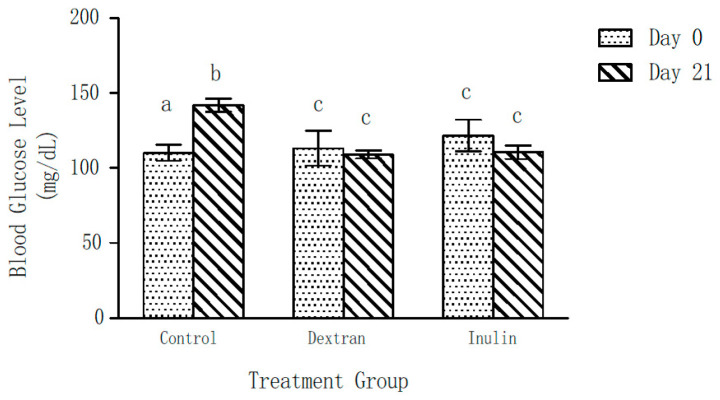
Effect of inulin and dextran type polysaccharides on blood glucose level of mice. The error bars represent standard error of the mean (SEM). ^a^
*p* < 0.05, compared with the blood glucose level at Day 21 within the control group ^b^
*p* < 0.05, compared with the blood glucose level at Day 0 within the control group. ^c^
*p* < 0.05, blood glucose levels of the dextran- and inulin-fed groups compared with the control group at Day 21.

**Figure 4 ijms-24-05314-f004:**
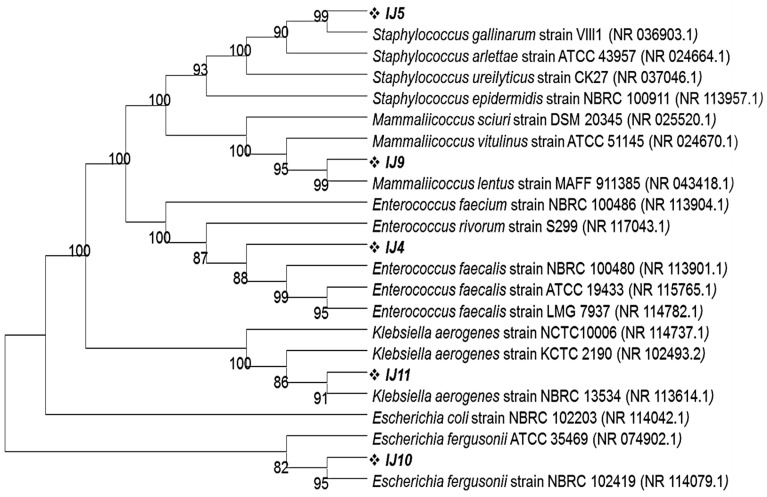
Phylogenetic tree showing the clustering of mice intestinal pathogens grown on CHROMagar plates on the basis of 16S rRNA gene sequences with their closely related homologs. The phylogenetic relationship was inferred using the neighbor-joining method by MEGA11 software program. The bootstrap consensus tree was inferred from 100 replicates.

**Figure 5 ijms-24-05314-f005:**
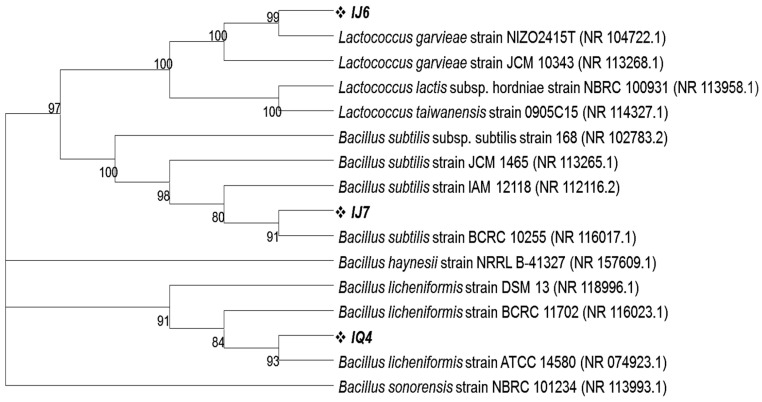
Phylogenetic tree showing the clustering of intestinal bacteria grown on MRS plates on the basis of 16S rRNA gene sequences with their closely related homologs. The phylogenetic relationship was inferred using the neighbor-joining method by MEGA11 software program. The bootstrap consensus tree was inferred from 100 replicates.

**Figure 6 ijms-24-05314-f006:**
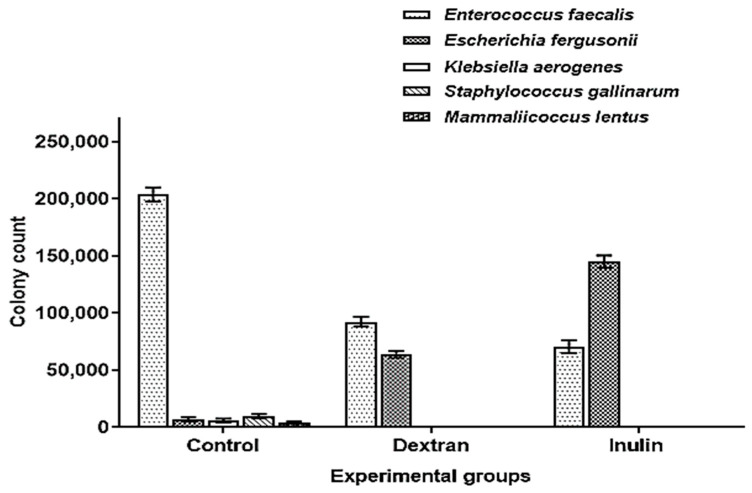
Comparative analysis of intestinal microbiota in different treatment groups of mice.

**Table 1 ijms-24-05314-t001:** Effect of dextran and inulin on total serum cholesterol and fecal lipid levels of mice.

Diet Group	Total Cholesterol (mg/dL)	Total Fecal Lipids (mg/g of Fecal Material)
Control	175.2 ± 6.4	11.0 ± 0.52
Dextran	135.5 ± 6.7	11.2 ± 0.48
Inulin	151.7 ± 5.1	19.6 ± 1.05

**Table 2 ijms-24-05314-t002:** Comparison of organ weights of mice from different diet groups.

Experimental Group	Relative Weight of Organs in Grams					
	Liver	Kidneys	Pancreas	Lungs	Heart	Stomach and Spleen
Control Group	1.70 ± 0.05	0.21 ± 0.01	0.21 ± 0.01	0.30 ± 0.02	0.14 ± 0.01	0.70 ± 0.03
Dextran Group	1.65 ± 0.09	0.20 ± 0.02	0.20 ± 0.02	0.28 ± 0.01	0.14 ± 0.01	0.65 ± 0.05
Inulin Group	1.70 ± 0.06	0.19 ± 0.01	0.18 ± 0.02	0.28 ± 0.02	0.15 ± 0.01	0.74 ± 0.04

The values are given as mean ± SEM. Each mean value represents 6 replicates of mice samples (*N* = 6).

**Table 3 ijms-24-05314-t003:** List and 16S rRNA gene sequence-based identification of bacterial isolates from mice intestine obtained on CHROMagar and MRS-agar media.

Code of Isolate	NCBI Accession Number	Colony Color on Chromagar	Colony Morphology	16S rRNA Gene Based Homologous Strain Prdicted from Blastn Result	Identity %
IJ4	OM049231	Turquoise blue	diplococci, non motile	*Enterococcus faecalis* strain NBRC 100481	100
IJ5	OM049232	Golden opaque	Small Round cells, non motile	*Staphylococcus gallinarum* strain VIII1	99.80
IJ9	OM049236	Light blue	Small Round cells, non motile	*Mammaliicoccus lentus* strain MAFF 911385	99.90
IJ10	OM049237	Pink	Non motile rods	*Escherichia fergusonii* strain NBRC 102419	99.49
IJ11	OM049238	Metallic Blue	Rods	*Klebsiella aerogenes* strain NBRC 13534	99.68
IJ6	OM049233	White transparent	non motile cocci, mostly diplococci	*Lactococcus garvieae* strain NIZO2415T	98.11
IJ7	OM049234	Milky white	Motile rods	*Bacillus subtilis strain* BCRC 10255	99.80
IQ4	ON909763	Milky white	Motile rods	*Bacillus licheniformis* strain ATCC 14580	98.64

**Table 4 ijms-24-05314-t004:** Effect of dextran and inulin supplemented diet on intestinal microbiota of mice.

Microorganisms	Treatment Groups *			Significant Difference (*p* Values < 0.05)		
	10^5^ × CFU/g			Control vs.		Dextran vs. Inulin
	Control	Dextran	Inulin	Dextran	Inulin	
*E. faecalis*	2.10 ± 0.07	0.92 ± 0.02	0.73 ± 0.05	Yes	Yes	No
*K. aerogenes*	0.058 ± 0.009	ND	ND	No	No	No
*S. gallinarum*	0.096 ± 0.01	ND	ND	No	No	No
*E. fergusonii*	0.06 ± 0.01	0.63 ± 0.01	1.43 ± 0.04	Yes	Yes	Yes
*M. lentus*	0.04 ± 0.005	ND	ND	No	No	No

* The values are given with ± SEM. Each mean value represents 6 replicates of samples from mice. ND: Not detected.

## Data Availability

The datasets generated during and/or analyzed during the current study are available from the corresponding author on reasonable request.

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
