# Peer review of "Dietary Supplementation of Microbial Dextran and Inulin Exerts Hypocholesterolemic Effects and Modulates Gut Microbiota in BALB/c Mice Models"

_ijms, 2023, doi:10.3390/ijms24065314_

Round 1

Reviewer 1 Report

The manuscript entitled “Effect of dietary supplementation of microbial dextran and inulin exopolysaccharides on metabolomics and microbiome of BALB/c mice models” by  Jaward et al gives an overview on the effect of microbial polysaccharides dextran- and inulin on microbiome and metabolome in mice.

The manuscript is well written and will be interesting for the readers of IJMS however few points should be addressed.

Figure 1, numbers interfere with figure

The quality of phylogenetic tree is poor

References need update.

Conclusion should be added separately.

Author Response

Response to Reviewer(s)' Comments to Author

Dear Editor and Reviewers,

Thank you very much for reviewing our article (Manuscript ID ijms-2192626) and providing very useful comments for its further improvement. We have carefully checked the article and revised the article according to your comments. The revised parts have been highlighted in track changes and detailed in a pointwise manner as below.

Reviewer 1 Comments to the Author

The manuscript entitled “Effect of dietary supplementation of microbial dextran and inulin exopolysaccharides on metabolomics and microbiome of BALB/c mice models” by Jaward et al gives an overview on the effect of microbial polysaccharides dextran- and inulin on microbiome and metabolome in mice.

The manuscript is well written and will be interesting for the readers of IJMS however few points should be addressed.

Comment: Figure 1, numbers interfere with figure

Response: The comment has been addressed and the figure is improved. The numbers are now added below Figure 1 in the manuscript to avoid interference.

Comment: The quality of phylogenetic tree is poor

Response: The quality and resolution of the phylogenetic trees have been improved as suggested by the worthy reviewer.

Comment: References need update.

Response: All references have been re-checked for their relevant content cited in the manuscript.

Conclusion should be added separately.

Response: The conclusion section has been added separately in the manuscript (please see Section 5, Line 673-679), as per suggestion of the worthy reviewer.

Author Response

Reviewer 2 Comments to the Author

Comment: It is difficult to understand some of the things done and why. The manuscript has many typos, and the study findings cannot be valid without addressing the concerns below.

Response: Thank you for the valuable comments. The manuscript has now been improved as per your suggestions please.

Comment: It is unclear whether the mice were kept in separate cages, i.e., one mouse per cage.

 Response: The different groups of mice were placed in different cages. Each group had six mice and all the six mice of one group were placed together in one cage. A total of three cages were used in the experiment; one for a control group of mice, second for dextran-fed group of mice, and third for inulin-fed group of mice. The statement in the manuscript has been modified to make it more clear in Lines 581-582.

Comment: Did the authors balance the total body weight among the groups?

Response: The initial body weight of all groups of mice was almost similar before the experiment. the weight of each mouse was recorded at Day 0 and the changes in body weight were determined accordingly at the end of experiment. However, the total body weight was found significantly different at the end of the experiment among the control and prebiotics-treated groups. This explanation is given in the manuscript in Section 4.4, Line 609-610.

Comment: The authors should be specific about the amount of feed given to the mice per day and how much was consumed by each mouse.

Response: Thank you for this observation. As already mentioned, all the six mice for each group were placed together in one cage and were given free access to feed (ad libitum), it is rather difficult to explain the how much feed was consumed by each mouse. However, the feed was provided @ 10g/mice/day (so, total 60g feed per day per cage containing six mice). Collectively, about 80% of the provided feed was consumed by each group of mice.

 Comment: Will the sterilization of the feed at 121 oC not significantly alter the nutritional composition of the feed used for the study?

Response: Sterilization of the feed was performed for relatively shorter time (10 min), and the Control and Experimental groups were provided with the feed sterilized adopting the same procedure. Moreover, the prebiotic polysaccharides were added after the sterilization of feed. Therefore, sterilization may not have effect on the comparative data obtained from Control and Experimental animals. we followed the same protocol.

 Comment: Did the authors sterilize the feed every day? If not, how did they store the feed after sterilization?

Response: The feed was not sterilized every day but was sterilized and prepared twice a week. For this purpose, the feed was sterilized in beakers or glass jars which were stored at 4 °C without opening the lid. The prebiotic-added dietary formulations were prepared under sterilized conditions to avoid any contamination.

 Comment: Did the animals fast before euthanization

 Response: Yes, we were provided fasting to the animals for 20 hours before euthanization.

Comment: Did the authors combine the squeezed fluid for each group

Response: The squeezed fluid of each group was not combined but assessed individually for each animal for the analysis of the intestinal microbiota on chromagar, macCkoncky agar, and MRS agar. The statistical analysis was performed by taking data of all six mice for each group.

Comment: Change “scratched” to “scrapped.” If scrapped, what was used to scrape the intestines? The scrapped contents would be intestinal mucosa, right?

 Response: Thank you for this valuable comment. As per your suggestion, “scratched” has now been replaced with “scrapped” throughout the manuscript. The sterilized spatula was used to collect the scrapped fluid (Line 483). You are right, the scrapped content would be intestinal mucosa, and this thing has been mentioned in the manuscript now (Lines 210, 597, 640).

 Comment: Why were the faeces collected on the last day of the feeding trial?

 Response: As all the parameters studied in this study has been reported with the data taken after 21 days of feeding mice, the faecal samples were also collected on day 21 (last day) to make sure that measurable effects of feed with or without prebiotic EPSs has been occurred. 

Comment: How did the authors collect blood from the peritoneal cavity? Did they collect blood only or a combination of blood and fluid?

 Response: Thank you for pointing this unclear statement. As only blood was collected by heart puncture method, the statement has now been modified to make it more clear (Section 4.6, Lines 621-622).

Comment: It is unclear how the authors collected the sera from the venipuncture of the mice

Response: The method of serum collection of blood has been explained in section 4.6 of the manuscript in Lines 621-622. The blood collected by venipuncture was used as such for only glucose measurement, while blood serum was used for the determination of cholesterol concentration. This point has now been clarified in Line 632.

Comment: Did the authors make CHROM agar and MacConkey agar media separately or combined? If separately, what did they do with MacConkey agar media?

 Response: Both media were prepared separately. The MacConkey agar media was used to confirm the presence of Escherichia sp. in the intestinal fluid of mice which appeared in the form of pink colonies.

How did the authors make MRS agar?

 Response: The MRS agar was prepared by adding 2.5 % agar in the MRS media. The composition of MRS media is given in section 4.1 of the manuscript in Lines 548-551.

Round 2

Reviewer 2 Report

  1. “Change initial body weight of all groups of mice were almost similar” to “the initial weight among the groups were not significantly different…”
  2. Any scientific support for 20 hours of fasting? If yes, include the citation. Prolonged fasting can trigger a metabolic switch from glucose-based to ketone-based energy that may affect the results presented, including the gut microbiome.
  3. The collection of feces on the last day without food for 20 hrs may be inappropriate. The recommended fasting period is 12 – 14 hours.
  4. If the authors do not have valid responses to questions 2 & 3, they should add a study limitation section at the end of the manuscript.
  5. Recast the following statement in line 371 “…the blood glucose level of control group mice increased drastically with age.” Age was not a factor in the study.
  6. I suggest the deletion of metabolomics from the title to accurately reflect the focus of the study.
  7. There are some typos that the authors need to fix.

Author Response

  1. Comment: “Change initial body weight of all groups of mice were almost similar” to “the initial weight among the groups were not significantly different…”

Response: The change has been made according to the suggestion in line 113 of the manuscript.

 2. Comment: Any scientific support for 20 hours of fasting? If yes, include the citation. Prolonged fasting can trigger a metabolic switch from glucose-based to ketone-based energy that may affect the results presented, including the gut microbiome.

 Response: Thank you for this valuable suggestion. The fasting was provided not for exactly 20 hours but in between 16-20 h (Between 5 PM to 9 AM the next day, when euthanization was started). The food was removed in the evening and the euthanization was done the next morning. In some previous reports, fasting was provided for 16 -24 hours, as given in the following references:

  • Yamamoto, Y.; Takahashi, Y.; Kawano, M.; Iizuka, M.; Matsumoto, T.; Saeki, S.; Yamaguchi, H. In vitro digestibility and fermentability of levan and its hypocholesterolemic effects in rats. Nutr. Biochem. 1999, 10, 13-18.
  • Melo, F. C. B. C. D., Zaia, C. T. B. V., & Celligoi, M. A. P. C. (2012). Levan from Bacillus subtilis Natto: its effects in normal and in streptozotocin-diabetic rats. Brazilian Journal of Microbiology, 43, 1613-1619.
  • Jang, K. H., Kang, S. A., Cho, Y. H., Kim, Y. Y., Lee, Y. J., Hong, K. H., ... & Choue, R. W. (2003). Prebiotic properties of levan in rats. Journal of microbiology and biotechnology13(3), 348-353.

The reference has now been cited in the manuscript as per the suggestion of the worthy reviewer (please see Line 593).

  1. Comment: The collection of feces on the last day without food for 20 hrs may be inappropriate. The recommended fasting period is 12 – 14 hours.

Response: The bedding was removed 48 hours prior to euthanization so the collected fecal samples were of 2 days (the same procedure has been adopted by Yamamoto et al 1999, as referred above in Response to Comment 2).

  1. Comment: If the authors do not have valid responses to questions 2 & 3, they should add a study limitation section at the end of the manuscript.

Response: The 2nd and 3rd comments have been addressed.

  1. Comment: Recast the following statement in line 371 “…the blood glucose level of control group mice increased drastically with age.” Age was not a factor in the study.

 Response: The comment has been addressed as per suggestion.

  1. Comment: I suggest the deletion of metabolomics from the title to accurately reflect the focus of the study.

 Response: This comment is addressed and the title of the manuscript is modified according to the suggestion of the worthy reviewer.

  1. Comment: There are some typos that the authors need to fix.

Response: The typo errors have been corrected throughout the manuscript